# The Association between a MAOB Variable Number Tandem Repeat Polymorphism and Cocaine and Opiate Addictions in Polyconsumers

**DOI:** 10.3390/brainsci11101265

**Published:** 2021-09-24

**Authors:** César Mateu, Marta Rodríguez-Arias, Isis Gil-Miravet, Ana Benito, José M. Tomás, Gonzalo Haro

**Affiliations:** 1Unidad Salud Mental Burjassot, Hospital Arnau de Vilanova-Lliria, 46015 Valencia, Spain; cesmateu@gmail.com; 2Grupo de Investigación TXP, Departamento de Medicina, Cardenal Herrera University (CEU), 12006 Castellón de la Plana, Spain; isis.gil@uji.es (I.G.-M.); anabenitodel@hotmail.com (A.B.); 3Unidad de Investigación Psicobiología de las Drogodependencias, Departamento de Psicobiología, Facultad de Psicología, Universidad de Valencia, 46010 Valencia, Spain; Marta.Rodriguez@uv.es; 4Unitat Predepartamental de Medicina, Universitat Jaume I, 12071 Castellón de la Plana, Spain; 5Àrea de Psicobiologia, Universitat Jaume I, 12071 Castellón de la Plana, Spain; 6Unidad Salud Mental Torrent, Hospital General de Valencia, 46014 Valencia, Spain; 7Departamento de Metodología de las Ciencias del Comportamiento, Facultad de Psicología, Universidad de Valencia, 46010 Valencia, Spain; jose.m.tomas@uv.es; 8Psychiatry Department, Hospital Provincial de Castellón, 12002 Castellón de la Plana, Spain; 9School of Medicine, Cardenal Herrera University (CEU), 12006 Castellón de la Plana, Spain

**Keywords:** genetics, polymorphism, MAOB, cocaine, opioids, polydrug use

## Abstract

Genetic analysis of the association between alcohol, cocaine, and opiate addiction and variable number tandem repeat (VNTR) polymorphisms in monoamine oxidase B (MAOB) and serotonergic 5-hydroxytryptamine (serotonin) receptor 1B and 2C (HTR1B 21 and HTR2C) pathway genes was performed in a sample of 302 polyconsumers. Our genetic association analysis revealed a significant association between a 184 base pair (bp) VNTR polymorphism in the MAOB gene and addiction to cocaine and opiates. This work highlights new genetic marker associations in cocaine and opiate polyconsumer addictions. These data help to clarify and quantify the complex role of genetics in addictive disorders, as well as their future contribution to the prevention (genetic counselling), diagnosis (genetic diagnosis of vulnerability), and treatment (pharmacogenomics) of these disorders.

## 1. Introduction

Findings in the scientific literature are beginning to echo the differences seen between single substance consumers and subjects presenting a multiple or polydrug use pattern [1,2]. Some studies have shown that nearly half the subjects who come for treatment for dependency for one substance present consumption of other substances and share specific sociodemographic, personality, number of attempted suicide, and psychiatric comorbidity traits. Despite the high number of polyconsumer subjects who attend treatment, scientific work in relation to this population is very scarce [3,4,5].

There is solid evidence of the role that genetic factors play in the development of addictive disorders [6,7,8]. Recent studies using GWAS (Genome-wide association study) have identified several novel genes that are associated with cocaine and opioid use [9,10,11]. Marees et al., (2020) found 2976 novel candidate genetic loci for substance use traits, and identified genes and tissues through which these loci potentially exert their effects [12]. Other studies have observed correlations between different psychiatric disorders and substance use disorders [13]. Numerous brain circuits are altered in addiction, among them are the dopaminergic, serotoninergic pathways, and MAO, the metabolic enzyme of the two systems [14,15]. Thus, the genes in these pathways are ideal candidates for carrying out genetic association studies. For example, several studies have shown significant associations between addiction to a diverse range of substances and dopaminergic and serotoninergic pathway genes [16,17] but very few have shed light on addictions related to MAO genes except for the identification of some associations with alcohol [18]. 

In the serotoninergic pathway, the 5-hydroxytryptamine 1B receptor is a protein encoded by the *HTR1B* gene, located in the 6q13-15 genetic locus [19] and whose expression is highest in the striate region of the cingulate gyrus [20,21,22]. The relationship between this receptor and different psychiatric pathologies, such as depression or obsessive-compulsive disorder, is fairly well established [23,24]. In 1998 the first association between a polymorphism in this gene and alcohol addiction was identified, but this association was also produced in the simultaneous presence of antisocial behaviour and the results were relatively unreliable [19,25,26]. Positive associations have also been observed between this gene and alcohol dependence in the Taiwanese Han and Chinese populations [27,28], although some studies have been unable to replicate these results [29]. In the case of cocaine addiction, the only notable study is that carried out by Cigler et al. (2001), which, similar to a study on methamphetamine addiction carried out in a Japanese population [30], did not detect any association. With regard to opiates, a protective effect against heroin addiction has been observed in Caucasian subjects carrying the G allele of the A1180G SNP [31]. 

The 5-hydroxytryptamine (serotonin) 2C receptor gene (HTR2C: 5-hydroxytryptamine (serotonin) receptor 2C gene) is located in the Xq24 genetic locus and comprises 4–6 exons [32,33]. Some studies have related different polymorphisms in this gene to psychiatric pathologies. A decrease in the activity of this receptor has also been observed in suicide victims with a history of major depression [34], as well as an association between borderline personality disorder and a higher frequency of the G allele of the rs6318 SNP [35]. Regarding addiction, this SNP was initially related to alcohol addiction, but subsequent studies failed to show any relationship between rs6318 polymorphisms and alcoholism [36,37,38,39,40,41]. So far no human studies have been undertaken to relate *HTR2C* polymorphisms to cocaine or opiate addictions.

Studies on *MAOB* gene have mainly centred on changes in its enzymatic functionality in relation to tobacco addiction [42], Parkinson disease [43], and emotional regulation [44]. In a Spanish sample, the A644G SNP in intron 13 of *MAOB* has been related to an increased risk of schizophrenia, especially in women [45]. A VNTR (VNTR: variable number tandem repeat) (GT)n polymorphism in intron 2 of this gene has also been related to Parkinson disease, although not every study has confirmed this association [46,47]. A subsequent paper attempted, without success, to associate the same VNTR with anxiety and depression [48]. The fact that several studies have registered different *MAOB* activity levels in alcoholic and healthy subjects [49] has meant that *MAOB* levels are considered by some as a genetic marker for alcoholism, although the results are contradictory and highly discussed. Some studies point to low levels of this enzyme in impulsivity, sensation seeking, and type-II Cloninger alcoholism [49,50,51,52], while others did not find these associations [53,54,55,56,57]. 

Low *MAOB* activity has not only been related to type-II alcoholism, but is also present in alcohol addiction more generally, regardless of age of onset and the presence of antisocial behaviour [58]. Some studies have observed a transient increase in *MAOB* activity during early periods of alcohol withdrawal, with maximum activity being observed during the first and second week after the last ingestion [59]. However, no direct associations between specific *MAOB* polymorphisms and vulnerability to alcohol addiction have been observed [60], and we could find no research indicating any relationship between *MAOB* gene polymorphisms and cocaine or opiate addictions.

The majority of genetic association studies concentrate on specific substances but very few focus their attention on the polyconsumer population. Hence, our objective in this study was to perform a genetic association study in this population type, selecting genes that are not commonly studied in these subjects, such as the *HTR1B*, *HTR2C*, and *MAOB*.

## 2. Materials and Methods

### 2.1. Sampling and Phenotyping

The sample was obtained by consecutively recruiting patients who came to receive treatment either at a hospital level, in the Hospital Detoxification Unit at the Hospital Clínico Universitario in Valencia, or at the walk-in level in the Addictive Behaviours Units at the San Marcelino and Padre Porta clinics in Valencia and the Dual Pathology Program at the Hospital de La Ribera (Alzira, Valencia). The study protocol was approved by the ethics committee at Hospital Clínico Universitario and all participants provided signed consent for their participation in the study, which was performed in accordance with the Declaration of Helsinki. Exclusion criteria were the presentation of a pathology other than addiction, inability to read or write, and the presence of an intellectual deficit or marked psycho-organic deterioration that would have impeded the performance of psychometric tests. The patient sample comprised 303 Caucasian polyconsumers, 231 (76.24%) male and 72 (23.76%) female, aged 34.5 ± 8.01 years. The subjects were included as an addict if they had been diagnosed with abuse or dependence using the DSM-IV-TR criteria at any point of their life. In this study, subjects suffering, or who had suffered, an addiction to each of the main substances in the study could comorbidly present an addiction to one of the other principal substances; a subject that met the criteria for addiction to more than one substance was included in each of the applicable groups for addicts of each separate substance. To define the phenotype, the following psychometric tests were used: the European version of the Addiction Severity Index (EuropASI), the International Personality Disorders Examination [61], and the Mini International Neuropsychiatric Interview [62].

### 2.2. Genotyping

The DNA extraction process and sample genotyping was performed by following the protocol described by Freeman et al. (2003) [63]. The samples were mailed to the Institute of Psychiatry at King’s College London to be analysed. An epithelial cell sample was obtained for DNA extraction from a piece of cotton that the subjects rubbed on the inside of their cheeks for at least 30 s. The tubes containing the cotton samples were stored at room temperature, and their confidentiality was guaranteed up to the point they were mailed to the laboratory.

For all markers, the sense and antisense PCR primer reagents were combined equally to create a single reagent. All the primers in the PCR reaction were optimised to work in a single reaction. Each multiplex PCR reaction contained 9 µL of true allele PCR mix (PE Biosystems, Foster City, CA, USA), 13.5 pmol/µL of *MAOB* primer, 5.7 pmol/µL of *5HT1B* primer, 15.4 pmol/µL of *5HT2C* primer, and 25 ng of DNA. The multiplex conditions were as follows: initial denaturation at 95 °C for 4 min, followed by 30 cycles of denaturation at 95 °C for 1 min, annealing at 60 °C for 1 min, and extension at 72 °C for 1 min. This was completed by a final extension at 72 °C for 10 min.

Capillary electrophoresis of PCR products was performed on an ABI 3100 automated sequencer (PE Biosystems). The resulting data were analysed using Genemapper™ software, version 2.0 (PE Biosystems). A novel marker for *5HT1B* was identified using the Tandem Repeat Finder program [64]. Where possible, viable markers were selected and primers designed to amplify the repeating region; primers for the remaining genes were obtained from Konradi et al. (1992) for *MAOB*, and Yuan et al. (2000) for *5HT2C* [65,66]. The gene markers consisted of a (TG)n repeat 25,498 bp downstream of *5HT1B*, a (GT)n repeat in intron 2 of *MAOB*, and a (GT)n repeat in the 1027 bp promoter upstream of *5HT2C*, as described in Nash et al. (2005).

### 2.3. Statistical Analysis

Polymorphisms with a frequency of less than 10% were grouped into a “dummy” allele (allele 1000), whose results were not taken into account in the final analysis. Three dichotomous nominal variables were created for each of the main substances being studied, and we indicated whether or not each subject presented addiction to each given substance. The statistical analysis for genetic association was performed with contingency tables and chi-square tests between each polymorphism and the presence/absence of the addictions. To better interpret the contingency table results, we estimated the adjusted standardised residual [67]. Binary logistic regressions were also used to estimate the odds-ratio.

## 3. Results

The entire sample was made up of polyconsumers, since they all had addiction to at least two substances (not counting tobacco). At the time of evaluation 94.6% of the sample cohort were smokers, Tobacco was not related to any of the polymorphisms studied (HTR1B: χ^2^ = 9.565, *p* = 0.134; HTR2C: χ^2^ = 8.971, *p* = 0.062; MAOB: χ^2^ = 6.775, *p* = 0.561). With respect to the three main substances, 165 subjects (54.6%) presented alcohol addiction, 252 subjects (83.4%) presented cocaine addiction, and 223 subjects (73.8%) presented opiate addiction. Regarding other substances, 13.8% (*n* = 42) met the dependency criteria for cannabis, 21.4% (*n* = 65) for benzodiazepines, 0.3% (*n* = 1) for amphetamines, and the remaining 64.5% (*n* = 194) did not meet the dependency criteria for any additional substances.

With respect to the presence of personality disorders in the selected sample cohort, 50.2% (*n* = 152) presented a personality disorder when evaluated. The most frequent disorder was borderline personality disorder representing 17.2% (*n* = 52), followed by non-specific (not otherwise specified) personality disorder at 15.5% (*n* = 47), antisocial personality disorder at 13.2% (*n* = 40), and paranoid personality disorder at 8.9% (*n* = 27); the remaining personality disorders presented much lower percentages (schizotypal 0.3%; histrionic 0.3%; dependent 1%; obsessive-compulsive 1%; narcissistic 1.7%; schizoid 2%; and avoidant 5.3%).

Regarding the genetic marker frequencies, Table 1 details the frequencies of the polymorphisms studied.

No correlation was found between 5HT receptor gene polymorphism and alcohol (HTR1B: χ^2^ = 6.382, *p* = 0.094; HTR2C: χ^2^ = 1.636, *p* = 0.441), cocaine (HTR1B: χ^2^ = 4.863, *p* = 0.182; HTR2C: χ^2^ = 3.862, *p* = 0.145) or opiate addiction (HTR1B: χ^2^ = 1.071, *p* = 0.784; HTR2C: χ^2^ = 1.588, *p* = 0.452). No relation was found between MAOB polymorphism and alcohol (χ^2^ = 1.055, *p* = 0.901). Of all the genetic association analyses carried out, only significant associations were found in the VNTR polymorphism which spanned 184 bp in the *MAOB* gene, which was associated with addiction to cocaine and opiates. This allele was more commonly found in subjects who presented addiction to cocaine compared to subjects without an addiction to cocaine (χ^2^ = 10.358, df = 4, *p* = 0.035; adjusted standardised residual = 2.5). Of the subjects carrying the 184 bp polymorphism, 90.8% presented addiction to cocaine, and binary logistic regression showed that this difference was statistically significant (χ^2^ = 7.089, *p* = 0.008). The percentage of explained variance was estimated by Cox & Snell and Nagelkerke formulae, and the estimates were 0.012 and 0.021, respectively. In addition, the predictor had a statistically significant odds-ratio of 2.19 (95% CI: 1.18–4.00). That is, a patient having the 184 bp *MAOB* polymorphism made it 2.19 times more likely that they would be diagnosed as a cocaine addict.

The 184 bp polymorphism was also over represented among opiate addicts (χ^2^ = 9.813, df = 4, *p* = 0.044; adjusted standardised residual = 2.7). Of the subjects carrying this polymorphism, 82.9% presented an addiction to opiates, and binary logistic regression showed that this difference was statistically significant (χ^2^ = 7.486, *p* = 0.006). The percentage of explained variance was estimated using the Cox & Snell and Nagelkerke formulae, and the estimates were 0.0013 and 0.0019, respectively. Moreover, the predictor had a statistically significant odds-ratio of 1.91 (95% CI: 1.17–3.11). In other words, having the 184 bp *MAOB* polymorphism made it 1.91 times more likely that a given patient would be diagnosed as an opioid addict.

## 4. Discussion

In this present study we found a higher frequency of the 184 bp *MAOB* VNTR (GT)n polymorphism in subjects with cocaine or opiate addictions compared to subjects with an alcohol addiction. Therefore, the *MAOB* gene can be treated as a previously unidentified target gene in polyconsumer subjects. This finding is novel because, until now, no studies have explored this genetic association specifically in the polyconsumer population. These results indicate that carriers of this polymorphism are approximately twice as likely to present cocaine or opiate addictions compared to polyconsumer subjects with an alcohol addiction. We therefore infer that this genetic marker may promote a specific genetic vulnerability both to cocaine and opiate addiction which becomes apparent and is concentrated in polyconsumer subject populations. Thus, this possibility should contribute to widening the debate which has been developing for some years about whether the heritability of addiction is general or specific for certain substances [68,69,70]. With respect to alcohol addiction, no significant associations were revealed.

To date, research on *MAOB* has primarily focused on studying its enzymatic activity in smokers [71] and during consumption and abstinence from alcohol [59]. However, genetic studies have not found any associations between *MAOB* gene polymorphisms and addiction. A study that analyzed a *MAOB* (A/G) SNP polymorphism found no association with type-II Cloninger alcoholism [60]. However, it should be noted that past studies trying to find associations between *MAOB* and alcohol addiction were based on samples obtained from subjects addicted only to alcohol but who did present other comorbid pathologies such as personality disorders, post-traumatic stress disorder, and anxiety/depression spectrum disorders [60]. In contrast, the population selected in this present work mainly comprised polyconsumers and reflected the clinical population that normally come to specialised units for treatment.

Moreover, we usually refer to polyconsumers in general, but specific differences among the polyconsumer population are studied very little. The few genetic studies that have mapped the genome in search of genetic associations with polydrug use have compared polyconsumers with healthy subjects or to those addicted to one substance, resulting in the identification of some chromosomal regions which might be related to the abuse of several substances, although none of these were located in the *MAOB* gene. However, these studies did not compare any possible differences in the genetic associations specifically among polyconsumers [72,73,74,75]. Therefore, in this present work we chose to compare polyconsumers not against healthy subjects, but rather, within the polyconsumer group itself. This is because our scientific goal was to find markers which can establish differences in vulnerability specifically within the polydrug use population. Our results show differences in a *MAOB* gene marker in polyconsumer subjects with an addiction to cocaine or opiates compared to polyconsumers with an alcohol addiction. Likewise, we found no significant associations between polyconsumer substance addiction and either of the two serotoninergic pathway genes (*HTR1B* and *HTR2C*) we analysed which may therefore indicate that the MAO pathway is more relevant to polyconsumer subject genetic vulnerability.

A limitation of our work is that we only studied specific polymorphisms in a polyconsumer population without comparing them to healthy age-matched control subjects, However, we consider that this could also be one of the study’s strengths. This type of study is very rare in contrast to the more usual studies that compare patients with addictions to healthy subjects, and so it provides original results not yet available elsewhere. Furthermore, the healthy controls, even if they were matched in age, would most likely differ in several sociodemographic and health variables of the polydrug users, which would make the comparison questionable. Secondly, although personality traits are strongly related to the heritability of certain addictions, as is the case with alcohol and antisocial personalities, we treated these data as purely descriptive in this study and did not include them in our genetic analysis statistical calculations because in themselves they represent sufficient material for an entire different study. Thus, one of the main problems of GWAS (Genome-wide association study) is that they tend to focus on associations between single nucleotide polymorphisms individually and leaving aside their combinatorial effects. For this reason, although GWAS are clearly the future of science and will clearly contribute to genetically explaining many diseases, associative studies are still necessary to be able to interrelate epigenetics with pathology. Looking ahead, longitudinal data can be used in GWAS to improve power for identification of genetic variants and environmental factors that influence complex traits, such as addiction, over time.

## 5. Conclusions

This work has improved the knowledge and understanding of the genetics of a very little studied polyconsumer population. In clinical practice, it is very common for subjects with multiple substance addictions to come for treatment. Therefore, much more analysis of the polyconsumer population and the differences among it is still required. This is because in general polyconsumers tend not to be grouped into their own specific category in addiction studies, therefore meaning that important and particular nuances in their phenotype may escape detection. This disserves one of the most common patient groups seeking treatment in addiction services, who are perhaps also those with the highest vulnerability both socially and genetically. Thus, analysis of the distinctive features of this patient cohort, both at the genotypic and phenotypic level, can help to implement effective treatments and to provide specialised healthcare focussed on the differential characteristics of polyconsumer subjects.

## Figures and Tables

**Table 1 brainsci-11-01265-t001:** This is a table of polymorphism frequencies.

HTR1B		HTR2C		MAOB	
bp	N(%)	bp	N(%)	bp	N(%)
302	385 (69.5)	259	189 (32.5)	180	80 (13.7)
304	70 (12.6)	265	362 (62.2)	182	121 (20.6)
308	66 (11.9)	1000	31 (5.3)	184	141 (24.1)
1000	33 (6)			186	182 (31.1)
				1000	62 (10.5)

Note: base pair (bp).

## Data Availability

Data sharing is not applicable to this article.

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
