# Peer review of "The Association between a MAOB Variable Number Tandem Repeat Polymorphism and Cocaine and Opiate Addictions in Polyconsumers"

_brainsci, 2021, doi:10.3390/brainsci11101265_

Round 1

Reviewer 1 Report

Monoamine oxidase is one of the major enzymes responsible for the degradation of neurotransmitters in the synapses of the brain. MAOA and MAOB genes are X-linked at Xp11.4–11.3 and produce two different forms of the enzyme. MAOA is present in catecholaminergic neurons, whereas MAOB is present in astrocytes and serotonergic neurons. Previous studies using platelet MAO enzyme levels found association between low MAO levels and alcoholism, schizophrenia, depression, manic depressive disorder, suicide, attention deficit hyperactivity disorder and risk-taking, sensation seeking or externalizing personality traits. Genetic factors play an important role in the development of addictive disorders. In the current study a genetic analysis was conducted of the association between alcohol, cocaine, and opiate addiction in a sample of 302 polyconsumers and variable number tandem repeat (VNTR) polymorphisms in MAOB and serotonergic 5-hydroxytryptamine (serotonin) receptor 1B and 2C (HTR1B and HTR2C) pathway genes. Their results show association between a 184 base pair (bp) VNTR polymorphism in the MAOB gene and addiction to cocaine and opiates. This short study adds to the recent genome-wide association studies that identified several novel genes that are associated with cocaine and opioid use.

Strength: This study generates new information about genetic marker associations in cocaine and opiate polyconsumer addictions and will potentially help in the prevention, genetic diagnosis, and pharmacogenoimcs of these disorders.

Limitation: Study is limited in scope by not using age-matched controls such as relatives.

Author Response

We appreciate the opinions of the reviewer. We have modified line 245 in the limitations section, to comply with that opinion.

Reviewer 2 Report

Mateu and colleagues have reported on the association between monoamine oxidase B polymorphism and cocaine and opiate but not alcohol addiction. This association appeared to be selective, as there was no association with genes encoding serotoninergic receptors. Overall, the manuscript is well written and the results are interesting. However, a few major and minor issues were noted.

Major:

  1. About 95% of the subjects were also smokers but no mention of the association between nicotine addiction and MAOB polymorphism was studied or discussed in the current study.
  2. The result of the 5HT1A and 2C receptor gene polymorphisms should also be described in detail in the Results section although no association was found.
  3. Line 46, MAO is an enzyme and not a pathway. It is OK to state that "among them are the dopaminergic and serotoninergic pathways and MAO, the metabolic enzyme of the two systems.
  4. Line 205, Is this variant found in subjects with both cocaine and opiate user disorders? If not, why the word polyconsumer is used?

Minor:

  1. Please change the first sentence of the abstract to "Genetic analysis of the association between alcohol, cocaine, and opiate addiction and variable number tandem repeat (VNTR) polymorphisms in monoamine oxidase B (MAOB) and serotonergic 5-hydroxytryptamine (serotonin) receptor 1B and 2C (HTR1B 21 and HTR2C) pathway genes was performed in a sample of 302 polyconsumers."
  2. Please delete one of the Introductions (repeated twice).
  3. Line 35, please change multiple consumption to cosumption.
  4. Line 70, Please delete "2002,"
  5. Line 77, please delete the word "in" before Parkinson's disease and emotional regulation.

Author Response

We appreciate your comments. We have done quasi-all the major and minor changes suggested:

Major:

  1. About 95% of the subjects were also smokers but no mention of the association between nicotine addiction and MAOB polymorphism was studied or discussed in the current study.

We have mentioned shortly the association between nicotine addiction and MAO polymorphism at the introduction section (lines 76-77) and the discussion section (lines 215-216) with the references:

42-Fowler, J.S.; Volkow, N.D.; Wang, G.J.; Pappas, N.; Logan, J.; MacGregor, R.; Alexoff, D.; Shea, C.; Schlyer, D.; Wolf, A.P.; et al. Inhibition of monoamine oxidase B in the brains of smokers. Nature 1996, 379, 733–736, doi:10.1038/379733a0.

71-Berlin, I.; Anthenelli, R.M. Monoamine oxidases and tobacco smoking. Int. J. Neuropsychopharmacol. 2001, 4, 33–42.

However, we agree that it could have been taken into account in the statistical analyses, so we have added it as a limitation, lines 252-253: “It can also be considered as a limitation that although the majority of the patients were tobacco smokers, this addiction was not considered in the statistical analyses.”

  1. The result of the 5HT1A and 2C receptor gene polymorphisms should also be described in detail in the Results section although no association was found.

The polymorphism frequency of the 5HT1A and 2C receptor is shown at table 1. Currently it is not possible to provide more details, taking into account the deadline requested by the editors.

  1. Line 46, MAO is an enzyme and not a pathway. It is OK to state that "among them are the dopaminergic and serotoninergic pathways and MAO, the metabolic enzyme of the two systems.

We have made these clarifying changes in lines 56 and 57, according to the numbering received in the latest version of the manuscript, and following your suggestions.

  1. Line 205, Is this variant found in subjects with both cocaine and opiate user disorders? If not, why the word polyconsumer is used?

We have added the sentence “The entire sample was made up of polyconsumers, since they all had addiction to at least two substances (not counting tobacco).” at the beginning of the results section to clarify it (lines 176-177). Also, we have removed the word polyconsumer in line 201 to avoid confusions.

Minor:

  1. Please change the first sentence of the abstract to "Genetic analysis of the association between alcohol, cocaine, and opiate addiction and variable number tandem repeat (VNTR) polymorphisms in monoamine oxidase B (MAOB) and serotonergic 5-hydroxytryptamine (serotonin) receptor 1B and 2C (HTR1B 21 and HTR2C) pathway genes was performed in a sample of 302 polyconsumers."

We have changed the first sentence of the abstract, the new one is clearer. Thank you for the suggestion.

  1. Please delete one of the Introductions (repeated twice).

Done, thanks.

  1. Line 35, please change multiple consumption to cosumption.

It was redundant, we have changed it.

  1. Line 70, Please delete "2002,"

We think you are referring to “2002” of line 80 at the last version we received, we have removed it, thank you.

  1. Line 77, please delete the word "in" before Parkinson's disease and emotional regulation.

We think you are referring to line 87, we have removed both “in” as suggested.

Round 2

Reviewer 1 Report

The authors have addressed the concerns. The limitation, however, remains in not using appropriate controls.

Author Response

We have modified the text recognizing the limitation: A limitation of our work is that we only studied specific polymorphisms in a polyconsumer population without comparing them to healthy age-matched control sub-jects.

Thanks a lot for the suggestion.

Reviewer 2 Report

The authors adequately addressed the concerns of the previous version. Regarding the description of the result of the 5HT receptor genes, the authors can simply write that we found no correlation between 5HT receptor gene polymorphism and opiate or cocaine addiction. I have no further comments or concerns.

Author Response

We have added in line 198 the sentence: No correlation was found between 5HT receptor gene polymorphism and opiate or cocaine addiction.

Thanks a lot for the suggestion.

This manuscript is a resubmission of an earlier submission. The following is a list of the peer review reports and author responses from that submission.